# Task-agnostic Lifelong Robot Learning with Retrieval-based Weighted Local Adaptation

## Abstract

A fundamental objective in intelligent robotics is to move towards lifelong learning robots that can learn to manipulate in unseen scenarios over time. However, continually learning new tasks and manipulation skills from demonstration would introduce catastrophic forgetting due to data distribution shifts. To mitigate the problem, we store a subset of demonstrations from previous tasks and utilize them in two manners: leveraging experience replay to retain learned skills and applying a novel Retrieval-based Local Adaptation technique to recover relevant knowledge. Besides, task boundaries and IDs are unavailable in scalable, real-world settings, our method enables a lifelong learning robot to perform effectively without relying on such information. We also incorporate a selective weighting mechanism to focus on the most "forgotten" action segment, ensuring effective skill recovery during adaptation. Experimental results across diverse manipulation tasks demonstrate that our framework provides a plug-and-play paradigm for lifelong learning, enhancing robot performance in open-ended, task-agnostic scenarios.

## 1 Introduction

Significant progress has been made in applying lifelong learning to domains such as computer vision (Huang et al., 2024; Du et al., 2024; Cai & Müller, 2023; Gurbuz et al., 2024; Mai et al., 2021; Singh et al., 2024) and natural language processing (Shi et al., 2024; Razdaibiedina et al., 2023; de Masson D'Autume et al., 2019; Biesialska et al., 2020). However, extending lifelong learning to robotics poses additional challenges, as robots must interact with the environment under sequential decision-making constraints. The high cost and complexity of physical interactions (Zhu et al., 2022; Du et al., 2023) limit the amount of available training data, making it critical to develop effective strategies to sustain robots' long-term performance (Thrun & Mitchell, 1995). Additionally, in realistic and scalable scenarios, lifelong learning robots must operate in a task-agnostic setting, where they are not provided with specific task boundaries or IDs for each new task. This further complicates the challenge, as robots must continually learn without knowing task distinctions.

In practical lifelong robot learning settings (Liu et al., 2024; 2023) — distinct from offline pre/post-training on large cross-embodied datasets — robots learn tasks sequentially, with limited access to past data due to on-device storage, bandwidth, and privacy constraints. This process often faces significant task distribution shifts, leading to catastrophic forgetting. While many approaches have been proposed to address this challenge (Wang et al., 2024a), they often rely on task boundaries or IDs, limiting their scalability in open-ended real-world scenarios (Koh et al., 2021).

To tackle these issues, we propose a task-agnostic, memory-based approach for lifelong learning of robotic manipulation skills from demonstrations. Noteworthy, "task-agnostic" does not mean the robot ignores the task context—it should be aware of language instructions and observations to fulfill the job; rather, it means the proposed algorithm effectively handles multiple continually encountered tasks and integrates new knowledge without relying on known task boundaries or IDs. Our method employs a compact storage memory $\mathcal{M}$ that holds a small set of previous tasks' demonstrations. During training, we replay samples from $\mathcal{M}$ to preserve prior knowledge and skills. However, partial forgetting remains inevitable due to the multitasking nature of lifelong learning and the emphasis on training the current new task (Wang et al., 2024b). Therefore,

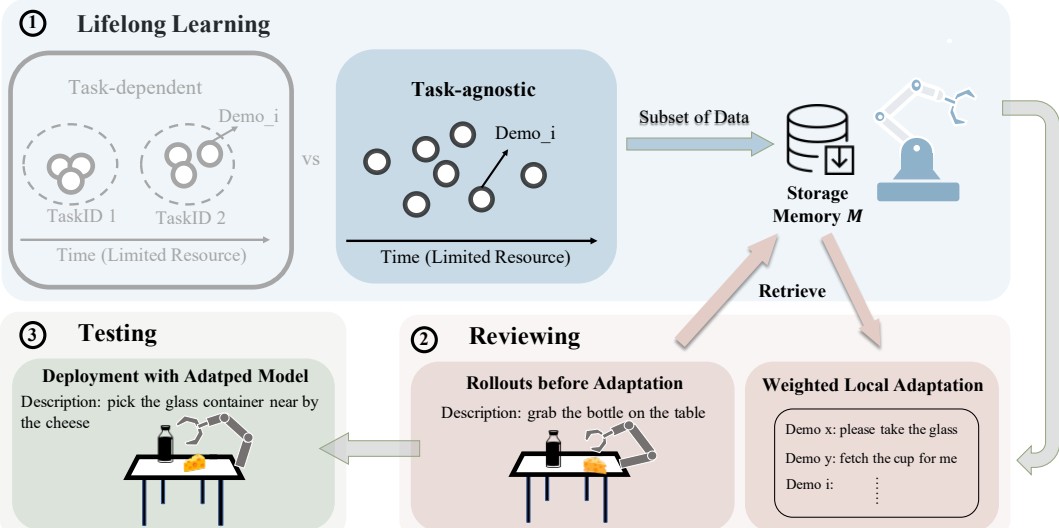

Figure 1: Method Overview. Our approach addresses the challenge of lifelong learning without relying on task boundaries or IDs. To emulate human learning patterns, we propose a method consisting of three phases: *Lifelong Learning*, *Reviewing*, and *Testing*. In the *Lifelong Learning* phase, the robot is exposed to various demonstrations, storing a subset of the data in a storage memory $\mathcal{M}$. During the *Reviewing* phase before policy deployment, the method retrieves the most relevant data to locally adapt the policy network, enhancing performance in the deployment scenario.

instead of attempting to retain every detail throughout continual training, more effective mechanisms are needed to mitigate forgetting.

Human studies show that once knowledge is learned, even if it is forgotten over time, an efficient and targeted review can trigger memory retrieval and rapidly recover lost proficiency—often faster than learning for the first time (Sara, 2000; Roediger & Butler, 2011; Ebbinghaus, 2013; MacLeod, 1988). Inspired by the findings, we enable robots to perform fast local adaptation before policy deployment, allowing them to review and recover forgotten manipulation skills accumulated through lifelong learning. Crucially, we use the same storage memory $\mathcal{M}$ for adaptation, avoiding any additional storage burden. Our system retrieves relevant demonstrations based on contextual similarity (Du et al., 2023; van Dijk et al., 2024; de Masson D'Autume et al., 2019) and automatically emphasizes the most failure-prone segments of each skill—typically critical steps where mistakes are likely to occur. This selective weighting, without requiring human intervention (Spencer et al., 2022; Mandlekar et al., 2020), promotes stable, task-agnostic lifelong learning. Our key contributions are summarized as:

- **Task-agnostic Retrieval-Based Local Adaptation:** A novel local adaptation strategy that retrieves relevant past demonstrations from $\mathcal{M}$ to recover forgotten skills, without requiring task boundaries or IDs.

- **Selective Weighting Mechanism:** An automated scheme that emphasizes the most challenging segments of retrieved demonstrations, optimizing real-time adaptation.

- **A General Paradigm Solution:** Our approach serves as a plug-and-play solution, complementing existing memory-based lifelong learning algorithms by enabling skill recovery in sequences of open-ended robotics tasks.

## 2 Related Works

**Lifelong Robot Learning.** Robots operating in continually changing environments need the ability to learn and adapt on-the-fly (Thrun, 1995; Grollman & Jenkins, 2007; Mendez-Mendez et al., 2023). In recent

years, lifelong robot learning has been applied to SLAM (Yin et al., 2023; Gao et al., 2022; Vödisch et al., 2022), navigation (Kim et al., 2024), and manipulation (Chen et al., 2023; Xie & Finn, 2022; Parakh et al., 2024; Lu et al., 2022; Auddy et al., 2023; Yang et al., 2022). Large language models have also been utilized to improve knowledge transfer (Bärmann et al., 2023; Tziafas & Kasaei, 2024; Wang et al., 2023).

To standardize the investigation of lifelong decision-making and bridge research gaps, Liu et al. (2024) introduced LIBERO, a benchmarking platform for lifelong robot manipulation where robots learn multiple atomic manipulation tasks sequentially. Recent works exploring lifelong robot learning based on it include Liu et al. (2023), which assigns a specific task identity to each task; Wan et al. (2024), which requires a pre-training phase to build an initial skill set before lifelong learning; and Lee et al. (2024), tackles multi-stage tasks by incrementally learning skill prototypes for each subgoal, which introduces additional complexities in managing subgoal sequences. However, catastrophic forgetting for lifelong robot learning remains an open challenge, especially when task boundaries and IDs are not available.

**Task-agnostic Lifelong Learning.** Despite the success of lifelong learning under clearly labeled task sequences, a significant gap remains in algorithms that can operate independently of task boundaries or IDs during both training and inference, thus aligning more closely with realistic and scalable scenarios. Many approaches (Lee et al., 2020; Chen et al., 2020; Ardywibowo et al., 2022) focus on specialized parameters via expanding network architectures. Meanwhile, researchers have tackled implicit task boundaries in regularization-based methods by consolidating knowledge upon detecting a loss "plateau" (Aljundi et al., 2019a; Kirkpatrick et al., 2017; Aljundi et al., 2018; Zenke et al., 2017) or through a bio-inspired approach using selective sparsity and recurrent lateral connections (Lässig et al., 2023). Memory-based algorithms further mitigate forgetting by prioritizing informative samples (Sun et al., 2022), discarding less critical examples (Koh et al., 2021), refining decision boundaries (Shim et al., 2021), or enhancing gradient diversity (Aljundi et al., 2019b). Methods aiming to exploit the replay buffer in online scenarios (Mai et al., 2021; Caccia et al., 2021) have also demonstrated notable success. However, these algorithms remain largely unexplored in robotic applications that entail sequential decision-making and real-world physical interactions.

**Robot Learning with Adaptation.** Recent advances have shown robots adapting to dynamic environments, such as executing agile flight in strong winds (O'Connell et al., 2022), adapting quadruped locomotion through test-time search (Peng et al., 2020), and generalizing manipulation skills from limited data (Julian et al., 2020; Memmel et al., 2024; Lin et al., 2024). To enable few-shot or one-shot adaptation, meta-learning has been extensively explored (Finn et al., 2017a) and successfully applied to robotics (Kaushik et al., 2020; Nagabandi et al., 2018; Finn et al., 2017b; Schmied et al., 2023). However, meta-learning methods typically assume access to a full distribution of tasks during meta-training, with both training and testing performed on tasks sampled from this distribution. In contrast, our lifelong robot learning scenario operating sequentially lacks such access, presenting unique challenges of catastrophic forgetting.

## 3 Preliminaries

To model realistic settings for lifelong robot learning, we define a set of manipulation tasks as $\mathbb{T} = \{\mathcal{T}_k\}, k = 1, 2, \ldots, T$, where each $\mathcal{T}_k$ encompasses a distribution over environmental variations $E_k$ (e.g., object positions, robot initial states) and language descriptions $G_k$ to guide robot's actions (e.g., "pick the bottle and put it into the basket," "place the bottle in the basket please"). From each $\mathcal{T}_k$, we sample specific environmental settings $e \sim E_k$ and language descriptions $g \sim G_k$ to generate a concrete scenario $\mathcal{S}_n^k \sim p(\mathcal{T}_k)$, which also serves as the basis for collecting demonstrations $\tau_n^k$. Multiple demonstrations form the training dataset $\mathcal{D}_k = \{\tau_n^k\}, n = 1, 2, \ldots, N$ for task $\mathcal{T}_k$.

Notably, multiple tasks may share overlapping distributions in either environmental settings or language descriptions. This natural setting closely mirrors real-world conditions, where it is difficult to determine which task generated a given scenario - tasks are not always divisible. This ambiguity underpins the proposed method's task-agnostic design, which emphasizes retrieving relevant information rather than relying on task boundaries or IDs.

The robot utilizes a visuomotor policy learned through behavior cloning to execute manipulation tasks by mapping sensory inputs and language description to motor actions. The policy is trained by minimizing the

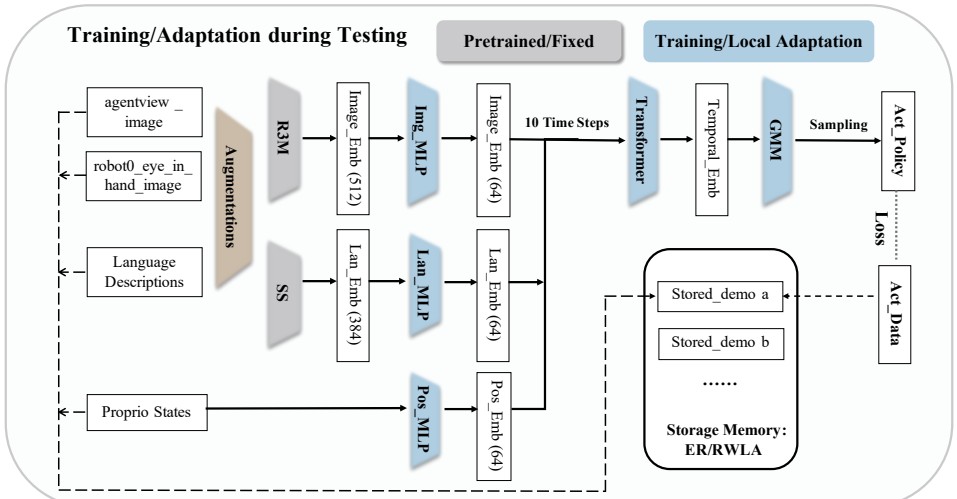

Figure 2: Policy Backbone Architecture used in Training and Testing. We input various data modalities into the system, including demonstration images, language descriptions, and the robot arm's proprioceptive input (joint and gripper states). Pretrained R3M (Nair et al., 2022) and (SentenceSimilarity, 2024) models process the image and language data respectively. Along with the proprioceptive states processed by an MLP, the embeddings are concatenated and passed through a Transformer to generate temporal embeddings. A GMM (Gaussian Mixture Model) is then used as the policy head to sample actions for the robot. Throughout both training and testing, we utilize a storage memory to store a subset of demonstrations gathered throughout the training process.

discrepancy between the predicted actions and expert actions from demonstrations. Specifically, we optimize the following loss function across a sequence of tasks $\mathbb{T}$ with $\mathcal{D}_k$. Notably, $\mathcal{D}_k$ is only partially accessible for $k < K$ from the storage memory $\mathcal{M}$, where $K$ denotes the current task:

$$\theta^* = \arg\min_\theta \frac{1}{K} \sum_{k=1}^K \mathbb{E}_{(o_t, a_t) \sim \mathcal{D}_k, \, g \sim G_k} \left[ \sum_{t=0}^{l_k} \mathcal{L} \left( \pi_\theta \left( o_{\leq t}, g \right), \, a_t \right) \right], \tag{1}$$

$\theta$ denotes the model parameters, $l_k$ represents the number of samples for task $k$, $o_{\leq t}$ denotes the sequence of observations up to time $t$ in demonstration $n$ (i.e., $o_{\leq t} = (o_0, o_1, \ldots, o_t)$), and $a_t$ is the expert action at time $t$. The policy output, $\pi_\theta(o_{\leq t}, g)$, is conditioned on both the observation sequence and the language description.

By optimizing this objective function, the policy effectively continues learning new knowledge and skills in its life span, without the need for task boundaries or IDs, thereby facilitating robust and adaptable task-agnostic lifelong learning.

## 4 Retrieval-based Weighted Local Adaptation (RWLA) for Lifelong Robot Learning

In this section, we outline our proposed method - Retrieval-based Weighted Local Adaptation (RWLA) - depicted in Figure 1, with corresponding pseudocode in Algorithm 1. To effectively interact with complex physical environments, the network integrates multiple input modalities, including visual inputs from workspace and wrist cameras, proprioceptive inputs of joint and gripper states, and language descriptions.

Instead of training all modules jointly in an end-to-end manner, we employ pretrained visual and language encoders that leverage prior semantic knowledge. Pretrained encoders enhance performance on downstream manipulation tasks (Liu et al., 2023) and are well-suited to differentiate between various scenarios and tasks. They produce consistent representations that are essential both for managing multiple tasks throughout lifelong training and for retrieving relevant data to support our proposed local adaptation before policy deployment.

---

**Algorithm 1** RWLA for Task-agnostic Lifelong Robot Learning

---

*Lifelong Learning* **Phase:**

    1. Initialize model parameter $\theta$, storage memory $\mathcal{M} = \{\}$, and tasks $\{\mathcal{T}_i\}$, $i = 1, 2, \ldots, T$

    2. $K \in \{1, 2, \ldots, T\}$

        (a) Train $\theta$ on $\mathcal{D}_K \cup \mathcal{M}$ using Eq 1

        (b) Randomly store a small number of demonstrations from $\mathcal{D}_K$ into $\mathcal{M}$

During deployment, robot encounters a testing scenario $\mathcal{S}_{deploy} \sim p(\mathcal{T}_i), 1 \le i \le T$:

*Reviewing* **Phase:**

    1. Rollout 10 episodes on $\mathcal{S}_{deploy}$ to assess robot's performance with $\theta$

    2. Retrieve $\tilde{N}$ demonstrations from $\mathcal{M}$ based on embedding distance using Eq 2 (Section 4.1)

    3. Compute $w_{t,n}$ based on selective weighting (Section 4.2.1)

    4. $\theta' \leftarrow$ Locally adapt $\theta$ using Eq 3 as skill recovery within limited epochs (Section 4.2.2)

*Testing* **Phase:** Test $\theta'$ in $\mathcal{S}_{deploy}$

---

When learning new tasks, the robot preserves previously acquired skills by replaying prior manipulation demonstrations stored in storage memory $\mathcal{M}$ (Chaudhry et al., 2019). Trained with the combined data from the latest scenarios and $\mathcal{M}$, the model can acquire new skills while mitigating catastrophic forgetting of old tasks, thereby maintaining a balance between stability and plasticity (Wang et al., 2024a). Figure 2 illustrates the network architecture, and implementation details are provided in Appendix A.2.

### 4.1 Data Retrieval

The proposed task-agnostic lifelong learning algorithm retrieves relevant demonstrations from $\mathcal{M}$ based on similarity to the deployment scenario $S_{deploy}$. Besides, due to the blurry task boundaries, some tasks share similar visual observations but differ in their task objectives, while others have similar goals but involve different backgrounds, objects, etc. To account for these variations, the retrieval process compares both visual inputs from the workspace camera (Du et al., 2023) and language descriptions (de Masson D'Autume et al., 2019) using $L_2$ distances of their embeddings, following a simple rule:

$$\mathcal{D}_R = \alpha_v \cdot \mathcal{D}_v + \alpha_l \cdot \mathcal{D}_l, \tag{2}$$

where $\mathcal{D}_R$ is the weighted retrieval distance, $\mathcal{D}_v$ represents the distance between the embeddings of the scene observation from the workspace camera, and $\mathcal{D}_l$ depicts the distance between the language description embeddings. The parameters $\alpha_v$ and $\alpha_l$ control the relative importance of visual and language-based distances. Based on the distances $\mathcal{D}_R$, the most relevant demonstrations can be retrieved from $\mathcal{M}$, as illustrated in Figure 3.

### 4.2 Weighted Local Adaptation

#### 4.2.1 Learn from Errors by Selective Weighting

To make the best use of the limited data, we enhance their utility by assigning weights to critical or vulnerable segments in each retrieved demonstration. Specifically, before local adaptation, the robot performs several rollouts on the encountered scenario using the current model trained during the *Lifelong Learning* phase. This procedure allows us to evaluate the model's performance and identify any forgetting effects (as illustrated in step 2, the *Reviewing* phase in Figure 1).

If a trial fails, we compare each image in the retrieved demonstrations against all images from the failed trajectories using $L_2$ distance of their embeddings. This comparison yields an Embedding Distance Matrix (EDM, shown in Figure 11) for each retrieved demonstration, where each value represents an embedding distance of a demonstration frame and an image from the failed rollout. This metric determines whether a particular frame has occurred during a failed rollout. Through this process, we identify the Separation Segment — frames in a demonstration where the failed rollout's behavior starts to diverge from the demonstration (see Figure 4). Since these Separation Segments highlight expected behaviors that did not occur, we consider

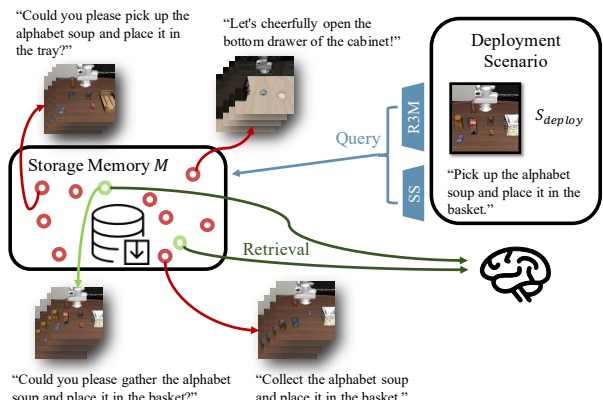

Figure 3: Storage memory $\mathcal{M}$ retains a small set of demonstrations during lifelong learning. To retrieve those most relevant to the current scenario, we compute a weighted distance (Eq 2) using image and language embeddings. Red and green circles indicate relevant and irrelevant demonstrations, respectively, which include language, visual observations, expert actions, and robot states. Retrieved demonstrations are then used for Weighted Local Adaptation.

Figure 4: Trajectory and Weighting Visualizations. To identify the point of failure, we compute the similarity between the retrieved demonstrations and failed trajectories at each frame. Once the separation segment is detected, higher weights are assigned to the frames in the segment of retrieved demonstrations during local adaptation.

them critical points contributing to failure. We assign higher weights to these frames, which will scale the losses during local adaptation. Detailed heuristics and implementation specifics are provided in Appendix A.4.

### 4.2.2 Local Adaptation with Fast Finetuning

Finally, we fine-tune the network's parameters to better adapt to the deployment scenario $S_{deploy}$ using the retrieved demonstrations from $\mathcal{M}$, focusing more on the Separation Segments identified through selective weighting. Despite this limited data, our experiments demonstrate that the model can effectively recover learned knowledge and skills and improve the robot's performance across various tasks. Overall, the proposed weighted local adaptation is formalized as follows:

$$\theta^* = \arg\min_\theta \sum_{n=1}^{\tilde{N}} \sum_{t=1}^{l_n} w_{t,n} \mathcal{L}\left(\pi_\theta(o_{\leq t,n}, g_n), a_{t,n}\right), \tag{3}$$

where $\tilde{N}$ is the number of retrieved demonstrations, $l_n$ is the length of demonstration $n$, and $w_{t,n}$ is the weight assigned to sample $t$ in demonstration $n$. The variables $o_{\leq t,n}$ and $a_{t,n}$ denote the sequence of observations up to time $t$ and the corresponding expert action, respectively, while $g_n$ is the language description for demonstration $n$. The parameter $\theta$ represents the network's parameters before local adaptation.

## 5 Experiments

We conduct a comprehensive set of experiments to evaluate the effectiveness of RWLA for task-agnostic lifelong robot learning. Specifically, our experiments aim to address the following key questions:

1. **Effect of Blurry Task Boundaries:** How do the blurry task boundaries influence the model's performance and data retrieval during testing?

2. **Advantages of RWLA:** Does the proposed approach enhance the robot's performance across diverse tasks?

3. **Impact of Selective Weighting:** Is selective weighting based on rollout failures effective?

4. **Generalizability:** Can our method be applied to different memory-based lifelong robot learning approaches, serving as a paradigm that enhances performance?

5. **Robustness:** How robust is our approach to imperfect demonstration retrieval, particularly when ambiguous task boundaries cause retrieved examples to mismatch the deploying scenario?

## 5.1 Experimental Setup

### 5.1.1 Benchmarks

We evaluate our proposed methods using LIBERO benchmarks (Liu et al., 2024): `libero_spatial`, `libero_object`, `libero_goal`, and `libero_different_scenes`. These environments feature a variety of task goals, objects, and layouts. The first three benchmarks all include 10 distinct task goals (e.g., "Put the bottle into the basket.", "Open the middle drawer of the cabinet."), each with up to 50 demonstrations collected from sampled simulation scenarios with different initial states of objects and the robot. Specifically, `libero_different_scenes` is created from LIBERO's `LIBERO_90`, which encompasses 20 tasks from distinct scenes.

We paraphrased the assigned single task goal into diverse language descriptions to obscure task boundaries (See Figure 7). These enriched language descriptions were generated by rephrasing the original task goal from the benchmark using a large language model provided by *Phi-3-mini-4k-instruct* Model (mini-4k instruct, 2024), ensuring consistent meanings while varying phraseology and syntax. Please see Appendix A.3 for more details.

### 5.1.2 Baselines

We evaluate our proposed method against the following baseline approaches:

1. **Elastic Weight Consolidation (EWC)** (Kirkpatrick et al., 2017): A regularization-based approach that relies on task boundaries and restricts network parameters' updates to prevent catastrophic forgetting of previously learned tasks.

2. **Experience Replay (ER)** (Chaudhry et al., 2019): A core component of our training setup, ER utilizes a storage memory to replay past demonstrations, helping the model maintain previously acquired skills and mitigate forgetting.

3. **Average Gradient Episodic Memory (AGEM)** (Hu et al., 2020): Employs a memory buffer to constrain gradients during the training of new tasks, ensuring that updates do not interfere with performance on earlier tasks.

4. **AGEM-RWLA**: An extension of AGEM that incorporates RWLA before policy deployment, enhancing the model's ability to adapt to specific scenarios. This allows us to assess the generalizability of our proposed method as a paradigm framework on other memory-based lifelong learning approaches.

5. **PackNet** (Mallya & Lazebnik, 2018): An architecture-based lifelong learning algorithm that iteratively prunes the network after training each task, preserving essential nodes while removing less critical connections to accommodate subsequent tasks. However, its pruning and post-training phases rely heavily on clearly defined task IDs, making PackNet a reference baseline when the IDs are well-defined.

### 5.1.3 Metrics

We focus on the success rate of task execution, as it is a crucial metric for manipulation tasks in interactive robotics. Consequently, we adopt the **Average Success Rate (ASR)** as our primary evaluation metric to address the challenge of catastrophic forgetting within the lifelong learning framework, evaluating success rates on three random seeds across all diverse tasks within each benchmark.

Table 1: Comparison with Baselines. The Average Success Rates (ASR, %) across various baselines are shown below. We provide PackNet's performance as a reference point for cases where task IDs are accessible. Both EWC and vanilla AGEM demonstrate weak performance across all benchmarks. Under our Retrieval-based Weighted Local Adaptation (RWLA) paradigm, both ER and AGEM show significant improvements over their vanilla counterparts, highlighting the effectiveness of RWLA.

| Benchmark\Method | Task Boundaries | Task IDs | Task-agnostic | | | |
|---|---|---|---|---|---|---|
| | EWC | PackNet | AGEM | AGEM-RWLA | ER | **ER-RWLA (ours)** |
| libero_spatial | 0.0 | 53.17 | 7.33 | 35.83 | 15.67 | **39.83** |
| libero_object | 1.50 | 73.67 | 27.17 | 51.17 | 56.50 | **62.33** |
| libero_goal | 0.33 | 66.33 | 10.83 | 58.67 | 52.33 | **62.33** |
| libero_different_scenes | 2.58 | 32.92 | 20.43 | 41.75 | 34.08 | **45.17** |
| Overall ASR | 1.10 | 56.52 | 16.44 | 46.85 | 39.65 | **52.42** |

### 5.1.4 Model, Training, and Evaluation

As illustrated in Figure 2, our model utilizes pretrained encoders for visual and language inputs: R3M (Nair et al., 2022) for visual encoding, Sentence Similarity model (SS Model) (SentenceSimilarity, 2024) for language embeddings, and a trainable MLP-based network to encode proprioceptive inputs. Embeddings from ten consecutive time steps are processed through a transformer-based temporal encoder, with the resulting output passed to a GMM-based policy head for action sampling. Specifically, R3M, a ResNet-based model trained on egocentric videos using contrastive learning, captures temporal dynamics and semantic features from scenes, while the Sentence Similarity Model captures semantic meanings in language descriptions, enabling the model to differentiate between various instructions.

To standardize the comparisons with baseline lifelong robot learning algorithms in LIBERO benchmarks, the model first undergoes a *Lifelong learning* phase, where it is trained sequentially on demonstrations from 10 or 20 tasks, depending on the specific benchmark, with each task trained for 50 epochs. A small number of demonstrations from each task is stored in $\mathcal{M}$, playing a dual-use for experience replay and RWLA. Every 10 epochs, we check the model's performance and save the version that achieves the highest Success Rate to prevent over-fitting.

After training on all tasks sequentially, we conduct *reviewing* and *testing* on various scenarios sampled from each task for comprehensive analysis. During the *reviewing* stage, we firstly evaluate potential forgetting by having the agent perform 10 rollout episodes on the deployment scenario $\mathcal{S}_{deploy}$. We then retrieve the most similar demonstrations from $\mathcal{M}$ and fine-tune the model for only 20 epochs using the retrieved demonstrations with selective weighting. Finally, we deploy the adapted model for 20 episodes—the *testing* phase—to assess performance improvements. All training, local adaptation, and testing in the benchmarks are conducted using three random seeds (1, 21, and 42) to reduce the impact of randomness.

### 5.2 Results

### 5.2.1 Comparison with Baselines

To address Question 2, we compared RWLA, with all baseline approaches. As shown in Table 1, ER-RWLA consistently outperforms baselines of EWC, AGEM, ER, and AGEM-RWLA. By incorporating local adaptation before policy deployment — our method mirrors how humans review and reinforce knowledge when it is partially forgotten — the continually learning robot could also regain its proficiency on previous tasks.

In contrast, PackNet — relying on explicit task IDs — allocates a separate slice of network weights to each new task. Noteworthy, this strategy works well early on, but as the number of tasks increases, the network's trainable capacity under PackNet diminishes, leaving less flexibility for future tasks. This limitation becomes evident in the libero_different_scenes benchmark, which includes 20 tasks, see Appendix A.6. PackNet's

success rate drops significantly for later tasks, resulting in poor overall performance and highlighting its constraints on plasticity compared with the proposed ER-RWLA.

Additionally, when we applied RWLA to the AGEM baseline (AGEM-RWLA), it also improved its performance, demonstrating the effectiveness of our method as a paradigm for memory-based lifelong robot learning methods. These findings support our conclusions regarding Question 4.

### 5.2.2 Ablation Studies

Table 2: Ablation Study on Selective Weighting. This table presents ASR (%) for uniform (RULA) and weighted (RWLA) local adaptation across 15, 20, and 25 epochs of adaptation under three random seeds, with evaluations conducted on all 10 tasks within the benchmarks: `libero_spatial`, `libero_object`, and `libero_goal`. Compared to RULA, selective weighting scheme improves the method's performance on most benchmarks.

| Benchmark | Method | 15 Epochs | 20 Epochs | 25 Epochs | Overall ASR |
|---|---|---|---|---|---|
| libero_spatial | *RULA* | 35.33 | 38.17 | **38.17** | 37.22 |
| | *RWLA* | **36.17** | **39.83** | 37.83 | **37.94** |
| libero_object | *RULA* | 57.83 | 60.67 | 58.00 | 58.83 |
| | *RWLA* | **58.00** | **62.33** | **61.50** | **60.61** |
| libero_goal | *RULA* | 61.33 | 62.00 | 66.17 | 63.17 |
| | *RWLA* | **62.83** | **62.33** | **67.50** | **64.22** |

We performed two ablation studies to validate the effectiveness of our implementation choices and address Questions 1, 3, and 5.

**Selective Weighting.** In the first ablation, we evaluated the impact of selective weighting on `libero_spatial`, `libero_object`, and `libero_goal` benchmarks to demonstrate its importance for effective local adaptation. We compared a variant of RWLA: RULA, which applies uniform local adaptation without selective weighting, adapting retrieved demonstrations uniformly. Both methods are trained with ER. Since early stopping during local adaptation at test time is infeasible, and training can be unstable, particularly regarding manipulation success rates, we conducted RWLA using three different numbers of epochs — 15, 20, and 25.

The results presented in Table 2 indicate that selective weighting enhances performance across different adaptation durations and various benchmarks, addressing Question 3. The gains over uniform adaptation are modest but systematic, echoing findings from (Byrd & Lipton, 2019), that importance weighting does not yield substantial performance jumps once expressive models have converged. Crucially, RWLA achieves these consistent improvements with negligible overhead — only a handful of demonstrations and a few extra adaptation epochs — making selective weighting a practical, low-cost boost for reliable skill recovery.

**Language Encoding Model.** To investigate the impact of language encoders under blurred task boundaries with paraphrased descriptions, we ablated the choice of language encoding model. Specifically, we compared our chosen Sentence Similarity (SS) Model, which excels at clustering semantically similar language descriptions, with BERT, the default language encoder from LIBERO. We selected the `libero_goal` benchmark for this study because its tasks are visually similar, making effective language embedding crucial for distinguishing tasks and aiding data retrieval for local adaptation.

Our experimental results yield the following observations:

(1) As illustrated in Figure 5 (a) and (b), the PCA results show that the SS Model effectively differentiates tasks, whereas BERT struggles, leading to inadequate task distinction. Consequently, as shown in Figure 5 (c), the model trained with BERT embeddings on `libero_goal` performs worse than the one trained with SS Model embeddings.

(2) Due to this limitation, BERT is unable to retrieve the most relevant demonstrations (those most similar to the current scenario from the storage memory $\mathcal{M}$). As a result, RWLA with BERT does not achieve good performance. These two findings address Question 1.

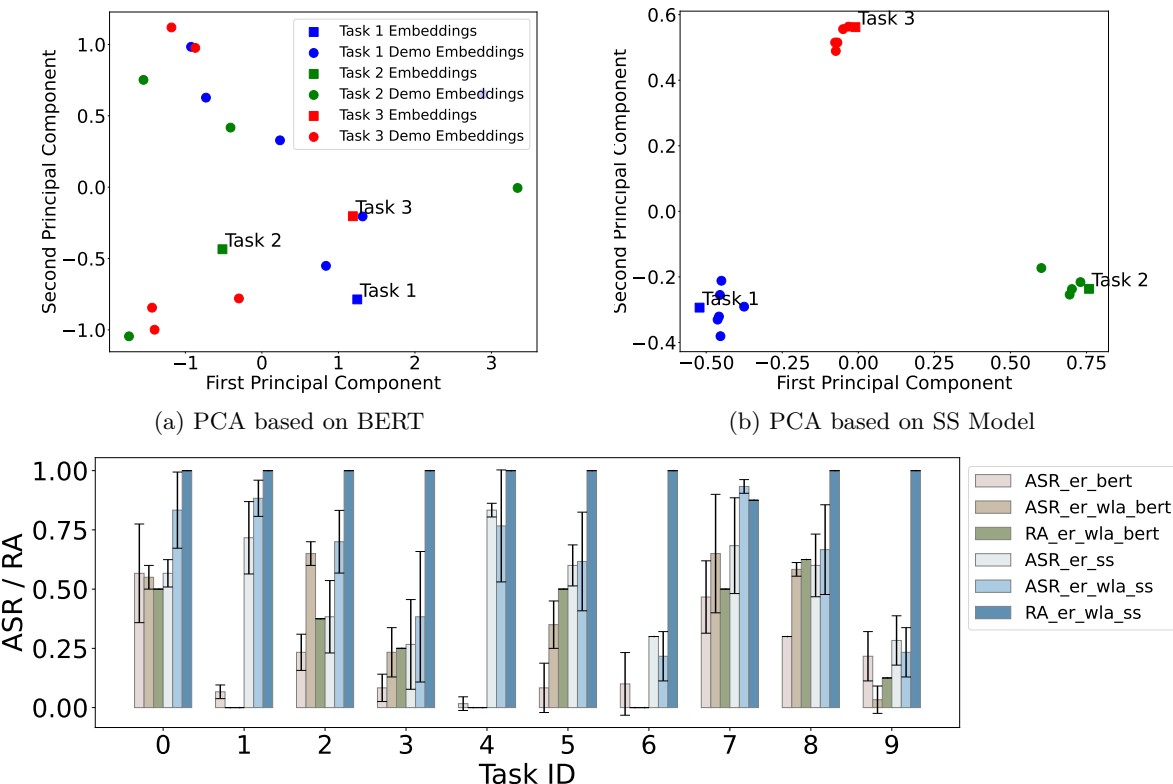

(a) PCA based on BERT           (b) PCA based on SS Model

(c) Bar Chart of Average Success Rates (ASR) and Retrieval Accuracy (RA) across 10 tasks

Figure 5: In Figures 5a and 5b, PCA is used to visualize the distribution of language embeddings of 3 tasks from BERT and SS, respectively. In Figure 5c, the SS model, which distinguishes task descriptions, has higher average success rates (ASR) and retrieval accuracy (RA) than BERT. The error bars represent the standard deviations of ASR and RA for each task over 20 repetitions with 3 random seeds.

(3) Interestingly, from Figure 5 (c), despite BERT's low Retrieval Accuracy (RA = proportion of the top $\tilde{N}$ demonstrations retrieved from $\mathcal{M}$ that come from the ground-truth task of $\mathcal{S}_{deploy}$), if it attains a moderately acceptable rate (e.g., 0.375), the RWLA based on BERT embeddings can still enhance model performance. This demonstrates the robustness and fault tolerance of our proposed approach, further addressing Questions 4 and 5.

# 6 Conclusion and Discussion

We introduce a task-agnostic lifelong robot learning framework that combines retrieval-based local adaptation with selective weighting before policy deployment. During continual learning, when performance on a previously learned task degrades, the robot triggers an on-demand "review" phase that uses just a few stored demonstrations to rapidly recover the forgotten skills — without requiring task IDs or boundaries. The method is robust and plug-and-play, seamlessly enhancing the memory-based lifelong learning robot's ability to retain and recover prior knowledge while it continues to acquire new skills.

A limitation of our framework is the scalability of the storage memory $\mathcal{M}$, as we continuously accumulate demonstrations. However, since image embeddings—serving dual purposes (input to the manipulation policy and data retrieval for local adaptation)—are generated by a pre-trained model, our approach is naturally extendable: this allows for significant storage reduction in future implementations, by simply storing smaller embeddings instead of raw images in $\mathcal{M}$.

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

# A Appendix for Task-agnostic Lifelong Robot Learning with Retrieval-based Weighted Local Adaptation

## A.1 Notations

Table 3: Mathematical Notations

| Symbol | Description |
|---|---|
| $k$ | Index of tasks, $k = 1, \ldots, K$ |
| $K$ | Total number of tasks |
| $n$ | Index of retrieved demonstrations |
| $\tilde{N}$ | Number of retrieved demonstrations |
| $i$ | Index of samples within a demonstration |
| $t$ | Time step |
| $l_k$ | Number of samples for task $k$ |
| $l_n$ | Length of retrieved demonstration $n$ |
| $\mathcal{T}_k$ | Task $k$ (represented by multiple goal descriptions) |
| $\mathcal{D}_k$ | Set of demonstrations for task $k$ |
| $\tau_i^k$ | Demonstration (trajectory) $i$ for task $k$ |
| $\mathcal{M}$ | Storage memory buffer |
| $o_t$ | Observation vector at time $t$ |
| $o_{\leq t}$ | Sequence of observation vectors up to time $t$ to deal with partial observability |
| $a_t$ | Action vector at time $t$ |
| $a_t^k$ | Action vector at time $t$ for task $k$ |
| $x_{i,n}$ | Input of sample $i$ in retrieved demonstration $n$ |
| $y_{i,n}$ | Label (action) of sample $i$ in retrieved demonstration $n$ |
| $\theta$ | Model parameters |
| $\theta^*$ | Optimal model parameters |
| $\theta_k$ | Model parameters after adaptation on task $k$ |
| $\pi_\theta$ | Policy parameterized by $\theta$ |
| $\pi_\theta(s_{\leq t}, \mathcal{T}_k)$ | Policy output given states up to time $t$ and task $\mathcal{T}_k$ |
| $\mathcal{L}$ | Loss function |
| $p(y \mid x; \theta)$ | Probability of label $y$ given input $x$ and parameters $\theta$ |
| $w_{i,n}$ | Weight assigned to sample $i$ in retrieved demonstration $n$ during adaptation |
| $\mathbb{E}$ | Expectation operator |
| $g_i$ | Goal descriptions in task $\mathcal{T}_k$ |

## A.2 Implementation and Training Details

### A.2.1 Network Architecture and Modularities

Table 4 summarizes the core components of our network architecture, while Table 5 details the input and output dimensions.

**Image Encoding:** We employ the R3M visual encoder (Nair et al., 2022), pretrained on large-scale robotic data, to extract rich image features. These features are then passed through a lightweight MLP that projects them into a compact 64-dimensional embedding.

**Language Encoding:** Natural language instructions are first embedded by a sentence-similarity model (SentenceSimilarity, 2024), yielding a 384-dimensional vector. A lightweight MLP then compresses this into a 64-dimensional latent representation, aligned with the other modalities.

Table 4: Network architecture of the proposed Model.

| Module | Configuration |
|---|---|
| Pretrained Image Encoder | ResNet-based R3M (Nair et al., 2022), output size: 512 |
| Image Embedding Layer | MLP, input size: 512, output size: 64 |
| Pretrained Language Encoder | Sentence Similarity (SS) Model (SentenceSimilarity, 2024), output size: 384 |
| Language Embedding Layer | MLP, input size: 384, output size: 64 |
| Extra Modality Encoder (Proprio) | MLP, input size: 9, output size: 64 |
| Temporal Position Encoding | sinusoidal positional encoding, input size: 64 |
| Temporal Transformer | heads: 6, sequence length: 10, dropout: 0.1, head output size: 64 |
| Policy Head (GMM) | modes: 5, input size: 64, output size: 7 |

Table 5: Inputs and Output Shape.

| Modularities | Shape |
|---|---|
| Image from Workspace Camera | $128 \times 128 \times 3$ |
| Image from Wrist Camera | $128 \times 128 \times 3$ |
| Max Word Length | 75 |
| Joint States | 7 |
| Gripper States | 2 |
| Action | 7 |

**Proprioceptive Encoding:** Joint angles and gripper states are processed through a small MLP stack, whose final layer also outputs a 64-dimensional vector. This ensures that vision, language, and proprioception share the same embedding size.

**Temporal Transformer:** At each timestep, we concatenate the three 64-dim modality embeddings and feed the sequence (up to 10 steps) into a 4-layer Transformer decoder. Each layer has 6 attention heads (64-dim head outputs), a 256-unit MLP, sinusoidal positional encodings, and 0.1 dropout to guard against overfitting.

**Policy Head:** The Transformer's final output is passed to a GMM-based head with 2 fully-connected layers. It predicts a 5-component Gaussian mixture—outputting per-mode means, standard deviations (clamped $\geq$ 1e-4), and mixture logits.

### A.2.2 Training Hyperparameters

Table 6 provides a summary of the essential hyperparameters used during training and local adaptation. The model training was conducted using a combination of **A40**, **A100**, and **L40S** GPUs in a multi-GPU configuration to optimize the training process. This distributed computing setup significantly enhanced efficiency, reducing the training time per benchmark from 12 hours on a single GPU to 6 hours using 3 GPUs in parallel. For each task, demonstration data was initially collected and provided by LIBERO benchmark. However, due to version discrepancies that introduced visual and physical variations in the simulation, we reran the demonstrations with the latest version to obtain updated observations. It is important to note that occasional rollout failures occurred because different versions of RoboMimic Simulation (Mandlekar et al., 2021) utilize varying versions of the MuJoCo Engine (Todorov et al., 2012).

Task performance was evaluated every 10 epochs using 20 parallel processes to maximize efficiency. The best-performing model from these evaluations was retained for subsequent tasks. After training on each task, we reassessed the model's performance across all previously encountered tasks. Although we report results only at the end of the full sequence, RWLA remains fully compatible with intermittent skill recovery during continuous deployment: it can be invoked when performance degradation happens, mirroring realistic lifelong-learning robots that adapt on an as-needed basis.

Table 6: Hyperparameter for Training and Adaptation.

| Hyperparameter | Value |
|---|---|
| Batch Size | 32 |
| Learning Rate | 0.0001 |
| Optimizer | AdamW |
| Betas | $[0.9, 0.999]$ |
| Weight Decay | 0.0001 |
| Gradient Clipping | 100 |
| Loss Scaling | 1.0 |
| Training Epochs | 50 |
| Image Augmentation | Translation, Color Jitter |
| Evaluation Frequency | Every 10 epochs |
| Number of Demos per Task | Up to 50 [1] |
| Number of Demos per Task in $\mathcal{M}$ ($\tilde{N}$) | 8 |
| Rollout Episodes before Adaptation | 10 |
| Distance weights $[\alpha_v, \alpha_l]$ for `libero_spatial` and `libero_object` | $[1.0, 0.5]$ |
| Distance weights $[\alpha_v, \alpha_l]$ for `libero_goal` | $[0.5, 1.0]$ |
| Distance weights $[\alpha_v, \alpha_l]$ for `libero_different_scenes` | $[1.0, 0.1]$ |
| Weights Added for Separation Segments | 0.3 |
| Clipping Range for Selective Weighting | 2 |
| Default Local Adaptation Epochs | 20 |

### A.2.3 Baseline Details

We follow the implementation of baselines and hyperparameters for individual algorithms from (Liu et al., 2024), maintaining the same backbone model and storage memory structure as in our approach. During training, we also apply the same learning hyperparameters outlined in Table 6.

### A.3 Details about Blurred Task Boundary setting

In this paper, we blur task boundaries by using multiple paraphrased descriptions that define the task goals. The following section elaborate more details about our dataset and process of task description paraphrase.

### A.3.1 Datasets Structure

Our dataset inherent the dataset from LIBERO Liu et al. (2024), maintaining all the attributes and data. Additionally, we add *demo description* to each demonstration to blur task boundary and augment language description during training (See Figure 6). Unlike the dataset from LIBERO, which groups demonstrations together under one specific task, our dataset wrap all demonstrations with random order to eliminate the task boundary.

### A.3.2 Description Paraphrase

We leverage the Phi-3-mini-4k-instruct model (mini-4k instruct, 2024) to paraphrase the task description. The process and prompts that we use are illustrated in Figure 7. As shown for the `libero_spatial` task in Figure 8, both BERT and Sentence Similarity Model struggle to distinguish tasks based on embeddings from the paraphrased descriptions. This observation further underscores the task-blurry setting in our experiments.

---

[1]For each task, demonstration data was collected from LIBERO, but due to differences in simulation versions, the demonstrations were rerun in the current simulation to collect new observations, with the possibility of occasional failures during rollout (see Subsection A.2.2 for details).

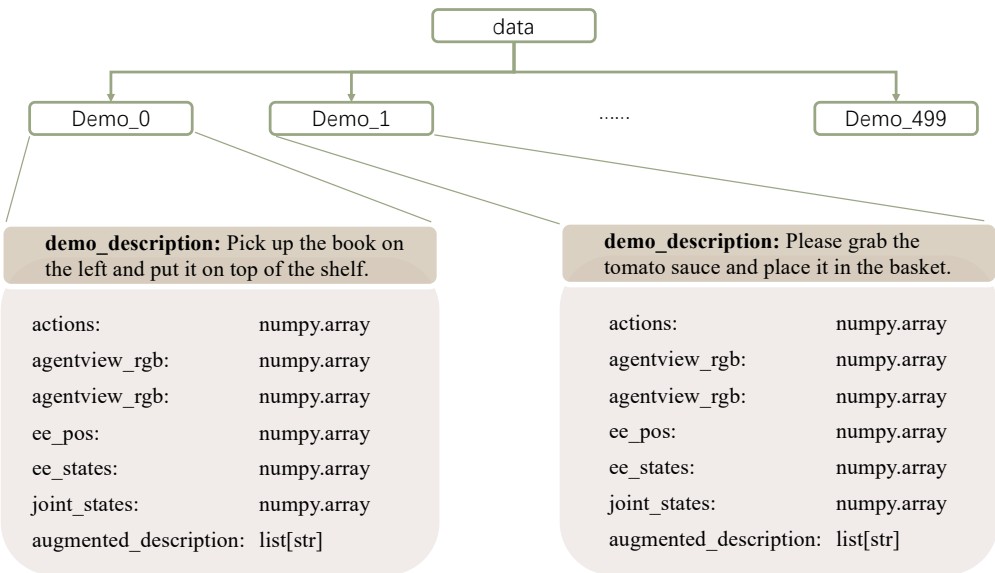

Figure 6: Data Structure

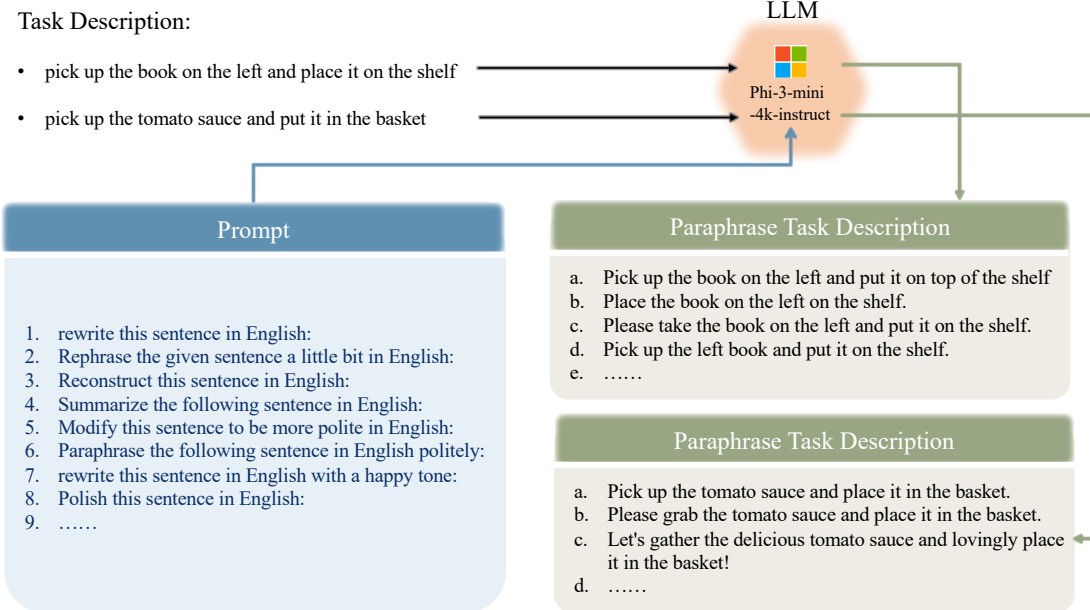

Figure 7: Paraphrase Description

## A.4 Detailed Heuristics and Implementations for Selective Weighting

Selective weighting strives to concentrate the adaptation loss on the *critical steps* of a demonstration—those at which the policy is most likely to diverge, forget, or fail. We detect these steps by measuring how a failed rollout departs from the demonstration in the image–embedding space.

For every retrieved demonstration we compute an **Embedding Distance Matrix** (EDM) between its frames and those of up to five failed rollouts. A row-wise minimum over the EDM yields the **Embedding Distance Curve** (EDC), which tracks, step-by-step, the smallest visual discrepancy seen so far. Because raw embedding distances are noisy (multi-modal actions, rendering noise), we smooth the EDC with a moving

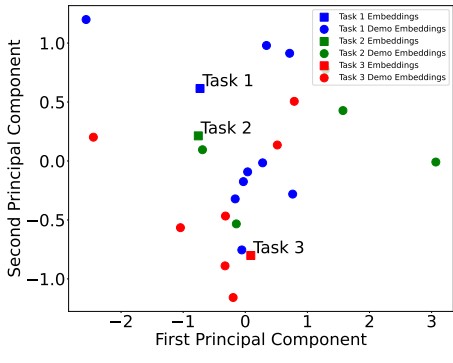

(a) PCA Visualization based on BERT

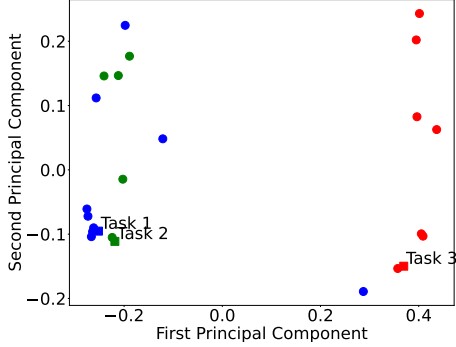

(b) PCA Visualization based on SS Model

Figure 8: Task Blurry Effect on `libero_spatial` benchmark. After paraphrasing the task descriptions, both BERT and SS models struggle to distinguish the tasks in `libero_spatial`.

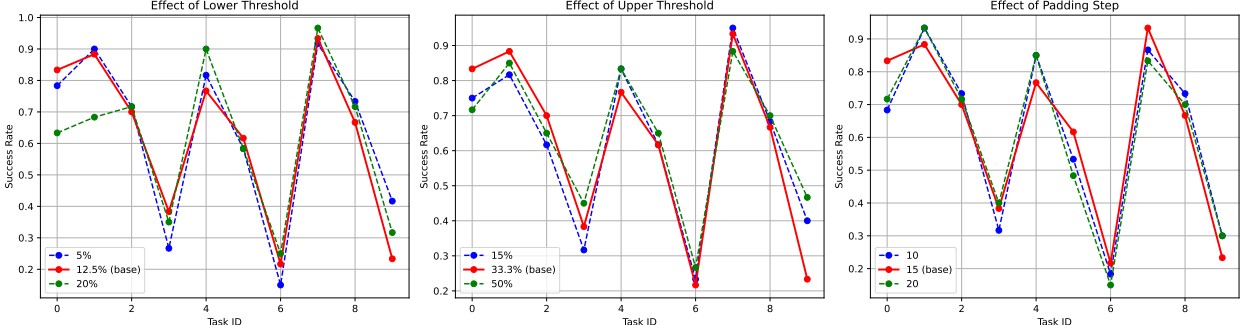

Figure 9: Hyperparameter Sensitivity Check.

average. The model typically fails when the EDC rises and stays high. We mark as a **Separation Segment** the last contiguous block of frames whose smoothed distance is between $\frac{1}{8}$ and $\frac{1}{3}$ of the curve's maximum. To further mitigate noise effects, we pad this block by $\pm 15$ frames.

Starting from a uniform weight vector, we add 0.3 to every frame inside the Separation Segment (for each retrieved demonstration based on the failed rollouts). Then, we clip weights to 2 and $\ell_1$-normalise all weights in each demonstration, producing $w_{t,n}$ used in Eq. equation 3. This focuses gradient updates on the vulnerable steps while keeping the overall loss scale stable.

Specifically, the hyperparameters such as thresholds $(\frac{1}{8}, \frac{1}{3})$ and padding (15 frames) were tuned on `libero_object` to best capture true divergence points; performance gains generalize across `libero_spatial` and `libero_goal` (Table 2).

### A.5 Additional Experiments

#### A.5.1 Comparison with Adaptation-based baseline

Our method can also be regarded as a form of test-time adaptation, akin to Schmied et al. (2023); Liu et al. (2023); Singh et al. (2024); Peng et al. (2020); de Masson D'Autume et al. (2019). To benchmark against an adaptation-oriented alternative, we chose **LoRA-FT** (Schmied et al., 2023). During the one-step adaptation phase, LoRA-FT (i) equips visual, language, and extra-modality encoders with LoRA adapters, (ii) inserts LoRA on the queries and values of every Transformer decoder layer, and (iii) fully fine-tunes the policy head.

A model with the identical architecture is first pre-trained for 50 epochs on the `libero_90` suite (90 short-horizon tasks) in a multitask fashion. We then adapt LoRA-FT to unseen tasks from `libero_spatial`, `libero_object`, and `libero_goal`. Both pre-training and adaptation use random seeds {1, 21, 42}. Unlike our ER-RWLA method—which has already experienced a lifelong-learning phase and therefore reviews only

a limited subset of demonstrations—LoRA-FT is granted access to the *entire* dataset of each target task during adaptation.

Figure 10 shows that our method consistently surpasses LoRA-FT across all three benchmarks. This aligns with the observation of Liu et al. (2024): pre-training on many short-horizon tasks can *hurt* downstream continual-learning or adaptation performance, and training from scratch may even outperform such pre-trained models. A likely cause is distributional mismatch—the 90 pre-training tasks differ substantially from the evaluation benchmarks—limiting positive transfer in LoRA-FT.

Note that, though we include such comparison with continual inference-time adaptation methods such as LoRA-FT, to assess our approach's performance relative to, our motivations and settings still differ greatly. Continual inference-time adaptation often aims to enable models to deal with continual domain transfer with streaming data, while our approach focuses on mitigating catastrophic forgetting through a brief, efficient "review" phase prior to policy deployment.

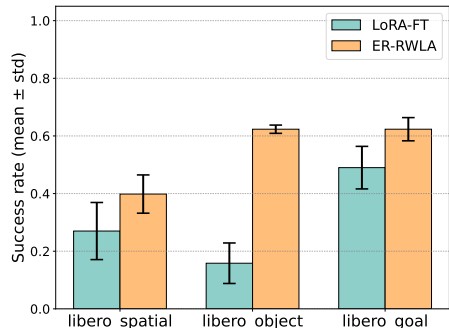

Figure 10: Average success rate ($\pm$ std) over three seeds on each benchmark. Both methods adapt for 20 epochs; LoRA-FT is allowed to see *all* demonstrations of the target task, whereas our proposed ER-RWLA only uses a small subset from $\mathcal{M}$.

### A.5.2 Hyperparameter Sensitivity Analysis on Selective Weighting.

Figure 9 presents the sensitivity analysis of the hyperparameters—lower threshold ($\frac{1}{8}$), higher threshold ($\frac{1}{3}$), and padding step (15 steps)—used to identify Separation Segments during selective weighting. The experiments are conducted on three random seeds as well. The results demonstrate that our proposed method's performance is robust to variations in these hyperparameters.

### A.6 Detailed Testing Results

We selected 20 typical scenarios among `libero_90`. The list of those scenarios can be found in Table 8. Additionally, the testing results of our method and baselines including **ER-RWLA**, **ER**, **Packnet**, are listed in Table 7.

### A.7 Further Discussions

#### A.7.1 Potential Forgetting during Local Adaptation

Our method addresses this issue through a robust deployment strategy. After sequential learning, we preserve the final model as a stable foundation. For each testing scenario, we fine-tune a copy of this model using our weighted local adaptation mechanism. Crucially, we always return to the preserved final model for subsequent scenarios, ensuring that each adaptation starts from the same well-trained baseline and previous

Table 7: Detailed Comparisons on `libero_different_scenes` Benchmark. It illustrates that after reaching the capacity of PackNet, its performance on new tasks would drop drastically.

| Task | ER-RWLA | ER | Packnet |
|------|---------|-----|---------|
| 0 | $0.85 \pm 0.08$ | $0.50 \pm 0.03$ | $1.00 \pm 0.00$ |
| 1 | $0.13 \pm 0.08$ | $0.27 \pm 0.06$ | $0.83 \pm 0.09$ |
| 2 | $0.73 \pm 0.09$ | $0.72 \pm 0.10$ | $0.92 \pm 0.02$ |
| 3 | $0.40 \pm 0.03$ | $0.13 \pm 0.02$ | $0.17 \pm 0.03$ |
| 4 | $0.93 \pm 0.04$ | $0.72 \pm 0.10$ | $1.00 \pm 0.00$ |
| 5 | $1.00 \pm 0.00$ | $0.57 \pm 0.16$ | $1.00 \pm 0.00$ |
| 6 | $0.52 \pm 0.04$ | $0.52 \pm 0.03$ | $0.78 \pm 0.04$ |
| 7 | $0.82 \pm 0.07$ | $0.63 \pm 0.09$ | $0.88 \pm 0.02$ |
| 8 | $0.32 \pm 0.07$ | $0.23 \pm 0.06$ | $0.00 \pm 0.00$ |
| 9 | $0.48 \pm 0.15$ | $0.38 \pm 0.12$ | $0.00 \pm 0.00$ |
| 10 | $0.23 \pm 0.06$ | $0.03 \pm 0.02$ | $0.00 \pm 0.00$ |
| 11 | $0.20 \pm 0.03$ | $0.10 \pm 0.06$ | $0.00 \pm 0.00$ |
| 12 | $0.23 \pm 0.09$ | $0.13 \pm 0.02$ | $0.00 \pm 0.00$ |
| 13 | $0.67 \pm 0.09$ | $0.83 \pm 0.04$ | $0.00 \pm 0.00$ |
| 14 | $0.15 \pm 0.03$ | $0.13 \pm 0.04$ | $0.00 \pm 0.00$ |
| 15 | $0.68 \pm 0.09$ | $0.30 \pm 0.08$ | $0.00 \pm 0.00$ |
| 16 | $0.03 \pm 0.03$ | $0.00 \pm 0.00$ | $0.00 \pm 0.00$ |
| 17 | $0.28 \pm 0.08$ | $0.02 \pm 0.02$ | $0.00 \pm 0.00$ |
| 18 | $0.10 \pm 0.08$ | $0.02 \pm 0.02$ | $0.00 \pm 0.00$ |
| 19 | $0.27 \pm 0.16$ | $0.58 \pm 0.07$ | $0.00 \pm 0.00$ |

Table 8: Selected Tasks for `libero_different_scenes` benchmark from `libero_90`

| Task ID | Initial Descriptions | Scenes |
|:---:|:---|:---:|
| 1 | Close the top drawer of the cabinet | Kitchen scene10 |
| 2 | Open the bottom drawer of the cabinet | Kitchen scene1 |
| 3 | Open the top drawer of the cabinet | Kitchen scene2 |
| 4 | Put the frying pan on the stove | Kitchen scene3 |
| 5 | Close the bottom drawer of the cabinet | Kitchen scene4 |
| 6 | Close the top drawer of the cabinet | Kitchen scene5 |
| 7 | Close the microwave | Kitchen scene6 |
| 8 | Open the microwave | Kitchen scene7 |
| 9 | Put the right moka pot on the stove | Kitchen scene8 |
| 10 | Put the frying pan on the cabinet shelf | Kitchen scene9 |
| 11 | Pick up the alphabet soup and put it in the basket | Living Room scene1 |
| 12 | Pick up the alphabet soup and put it in the basket | Living Room scene2 |
| 13 | Pick up the alphabet soup and put it in the tray | Living Room scene3 |
| 14 | Pick up the black bowl on the left and put it in the tray | Living Room scene4 |
| 15 | Put the red mug on the left plate | Living Room scene5 |
| 16 | Put the chocolate pudding to the left of the plate | Living Room scene6 |
| 17 | Pick up the book and place it in the front compartment of the caddy | Study scene1 |
| 18 | Pick up the book and place it in the back compartment of the caddy | Study scene2 |
| 19 | Pick up the book and place it in the front compartment of the caddy | Study scene3 |
| 20 | Pick up the book in the middle and place it on the cabinet shelf | Study scene4 |

adaptations do not influence future ones. This approach keeps local adaptations isolated and prevents the accumulation of forgetting effects.

### A.7.2 Future Work beyond LIBERO

Current LIBERO benchmarks treat similar scenarios (e.g., variations in object categories, layouts, or goals) as distinct tasks. However, this does not reflect how humans learn continually — we tend to generalize across such variations. Thus, we plan to identify equivariant features among similar tasks to avoid redundant retraining or adaptation. This strategy will further mitigate catastrophic forgetting and improve the system's plasticity in more complex lifelong robot manipulation tasks with the proposed RWLA algorithm.

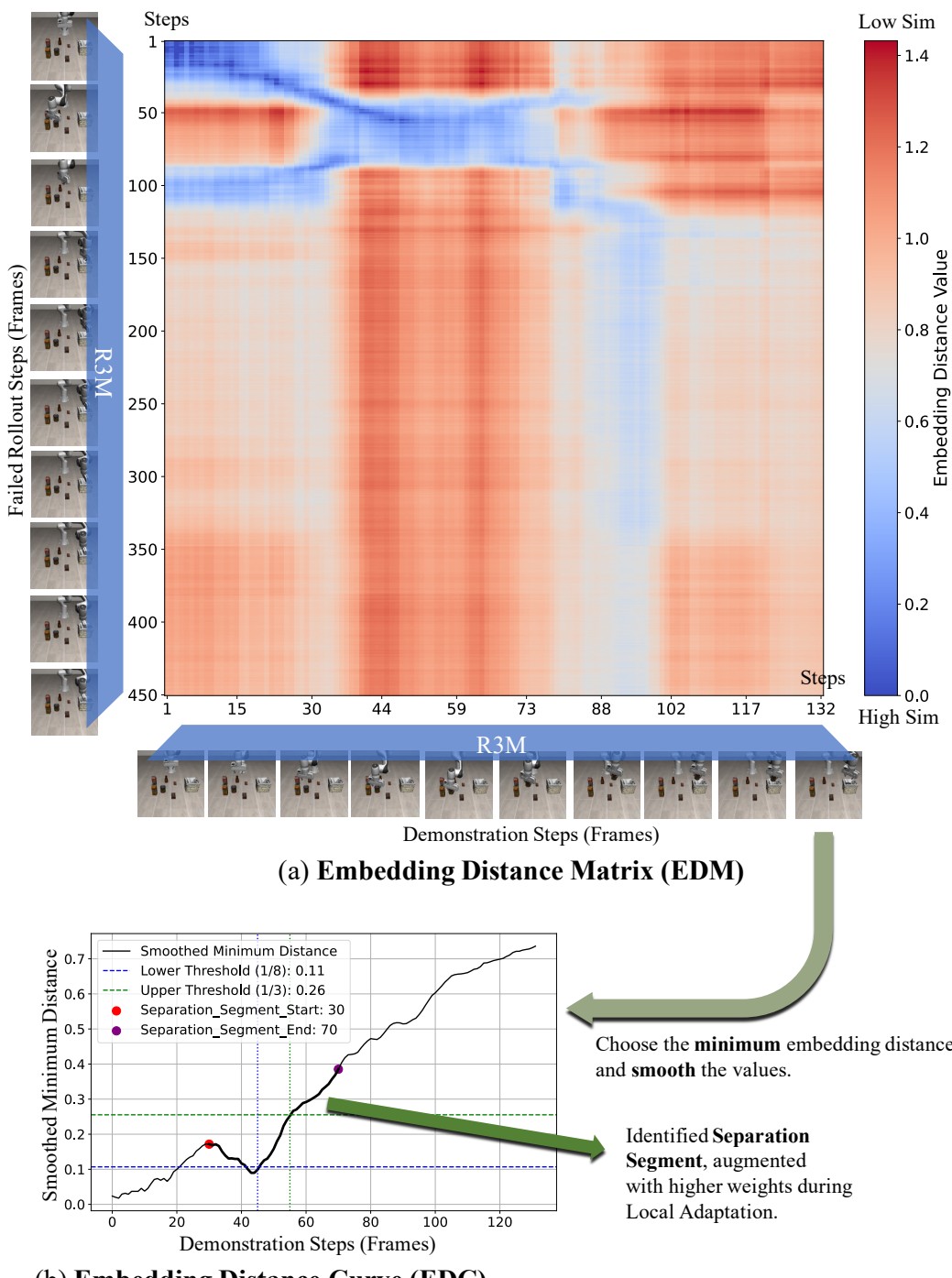

(a) **Embedding Distance Matrix (EDM)**

(b) **Embedding Distance Curve (EDC)**

Figure 11: Illustration of the selective weighting heuristic using (a) **Embedding Distance Matrix (EDM)** and (b) **Embedding Distance Curve (EDC)**. In the demonstration, the robot successfully picks up a jar and places it into a basket. In the failed rollout, the robot fails during the picking stage, resulting in the absence of subsequent steps. The steps surrounding the picking procedure are identified as the **Separation Segment** and are assigned higher weights during adaptation to address the model's shortcomings. Specifically, the Separation Segment is determined by the smoothed minimum $L_2$ distances from EDC—obtained from EDM, where each of its entry indicates the embedding distance between a demonstration and failed rollout frame, as shown in this figure.

