# OpenReview forum: "Task-agnostic Lifelong Robot Learning with Retrieval-based Weighted Local Adaptation"
_TMLR — Withdrawn by Authors_

### Review · Reviewer_drZp · 2025-11-02

**Summary Of Contributions:**

This paper proposes RWLA, a selective-training approach for adapting policies to seen but potentially forgotten scenarios after training them under the lifelong learning setting. RWLA builds on memory-based approaches that maintain a replay buffer that contains demonstrations for different tasks. When adapting to a testing task (seen but potentially in a slightly different scenario), RWLA first retrieves demonstrations from the buffer based on the similarity of the visual observations and language-based task descriptions to the current task. After data is retrieved, RWLA updates the policies using the same behavior loss but with a weighting that emphasizes the transitions in which the policies deviate from the demonstrations due to forgetting. Empirically, the paper shows the proposed approach achieves better learning performance than baselines that are unselective on retrieved demonstrations and transitions.

The strengths of the paper include
1. The proposed approach intuitively makes sense and (not surprisingly) appears to be quite effective.
2. The paper is well written and organized. Figures are very illustrative of the ideas presented. The experiment section is also clear with investigated research questions listed.

On the other hand, its main weaknesses are
1. The core mechanism is not precisely described in clear mathematical terms. Please see the Requested Changes for a list of key information to be added.
2. Insufficient description of the experimental protocol. Please see the Requested Changes for a list of key information to be added.
3. The reported empirical results are not rigorous. While only a small number of random (3) seeds are used in the experiments, statistical significance is not reported in the main results, including Tables 1 and 2.

**Audience:**

Yes

**Audience Explanation:**

Learning continually is an important topic for real-world applications and an interesting topic for research. This paper studies fast adaptation of policies trained in a continual setting. It implements an intuitive and natural approach, and it should be of interest to potential continual learning and robotics researchers.

**Broader Impact Concerns:**

The paper discusses methods for training physical robots continually. The current capability of robots to adapt continuously is still very limited and might not have an immediate significant societal impact.

**Claims And Evidence:**

No

**Claims Explanation:**

1. Vague descriptions of the approach and experiments without sufficient details stop the readers from fully understanding and reproducing the results. See Requested Changes for a list of missing information.
2. The empirical results lack statistical significance to support the claims (in Sections 5 and 6) about the effectiveness of the proposed method. Specifically, it’s unclear if the results in Tables 1 and 2 are statistically significant, as no relevant measure is provided.
3. This is a minor point and might need authors’ clarification: While the paper uses *task-agnostic* to describe the problem setting, it appears that the observable task description in language seems to be sufficient to play the role of task ID. This is supported by Figure 5(b), in which the language embedding of different instances of the same task is properly grouped. The reviewer is not sure if such a setting is considered task-agnostic in the relevant community. Either way, the authors should clarify and provide support for such a usage.

**Requested Changes:**

Changes that are crucial for a recommendation for acceptance:
1. Adding clear descriptions of missing algorithmic details:
  - What’s the definition of $\mathcal{L}$? Is it a simple L2 distance?
  - Is $S_{deploy}$ just one instance or a set of instances of a task? Should it be conditioned on $i$? How many instances are evaluated in the reviewing/testing phase in the experiments?
  - Precise definition of $\mathcal{D}_v$ and $\mathcal{D}_l$. How many transitions are used to calculate them? Are they averaged over some dimension?
2. Adding statistical measures for results in Tables 1 and 2. Without a statistical measure, it’s difficult to judge if a different set of random seeds would result in the same conclusions. If the results are not significant, the claims should also be weakened (whenever comparing different algorithms or variants).
3. Adding detailed information about the experiment protocol.
  - How many tasks are evaluated/tested after lifelong training?
  - If there are multiple tasks, is adaptation independent between tasks?
  - How many “similar demonstrations from $\mathcal{M}$” were retrieved?
  - What’s the criteria to keep data in $\mathcal{M}$ during (lifelong) training?
4. If proper, the learning curves during adaptation should also be added to demonstrate *fast* adaptation.

Minor questions or suggestions:
1. Should $l_k$ be a random variable instead of fixed per k?
2. In Section 5.1.1, I suggest just referring to LIBERO as one benchmark and referring to different settings *suites*, as in the original LIBERO paper. Otherwise, it might be easy to create confusion.
3. Figure 9 should also include uniform weighting as a baseline.

---

### Review · Reviewer_uzM9 · 2025-11-03

**Summary Of Contributions:**

### Summary
The paper introduces a task-agnostic lifelong learning framework for robotic manipulation that mitigates catastrophic forgetting. Extending memory-based approaches, this method proposes a Retrieval-based Local Adaptation procedure that, before deployment, retrieves context-relevant demonstrations from the replay to quickly restore forgotten skills. The approach includes a selective weighting mechanism that automatically identifies and emphasizes the most challenging segments during adaptation. Experiments demonstrate strong performance.

### Strengths
1. The manuscript is overall well-written and easy to follow. The related work is comprehensively summarized.
2. The idea of local adaptation before deployment is interesting.

### Weaknesses
1. Insufficient justification for problem setting
2. Missing descriptions of proposed approach
3. Insufficient empirical evidence

**Audience:**

No

**Audience Explanation:**

Please see the weaknesses in the **Problem Setting and Methodology** elaborated as follows:

I wonder how realistic it is to undergo a Reviewing Phase for each scenario S_{deploy} before deployment. In most continual learning problems, the model is tested on all benchmark tasks right after the training phase. The Review Phase requires additional training involving extra computational and time budget. Such a setting is not quite applicable to the broad range of real-world scenarios, especially highly complex robotics manipulation tasks.

In Algorithm 1 Lifelong Learning Phase, the procedure does not clearly specify how the training dataset D_K is constructed. The preliminaries describe such a dataset D_k contains up to N demonstrations per task T_k, where N=50 as in Table 6. Considering the failures (A.2.2), how do you meet this strict requirement (keeping rollout until reaching N, or rollout more than N and sample a subset, or otherwise)?

Could you provide more information on the temporal transformer? Why does each layer have positional encodings? What type of normalization is applied, if any?

Could you explain the heuristics or provide relevant citations for computing the coefficient w_{t,n}? Why adding 0.3, clipping to 2, and L1 normalization?

**Claims And Evidence:**

No

**Claims Explanation:**

Please see the weaknesses in the **Experiments** elaborated as follows:

It is critical to understand RWLA's standalone significance in the entire pipeline. Could you try the ablation setting that only optimizes the model in the reviewing phase, i.e., the "lifelong learning" phase collects demonstrations in the replay without optimizing parameters \theta, while the reviewing phase performs RWLA. I recommend experiments on different values for the replay capacity |M|, the number of demonstrations \tilde{N}, and the number of adaptation epochs (as in Table 2). Since RWLA is much less costly than the original lifelong learning phase, the proposed ablations should not require as much budget.

In the chosen benchmarks, either visual input or language description is already informative. To understand their individual importance, could you also ablate the hyperparameters \alpha_v \& \alpha_l by setting one of them to 0 in each experiment?

As evaluations are performed on 3 random seeds, could you also report the standard deviation for each success rate in Table 1 \& 2?

**Requested Changes:**

Please address the weaknesses above.

---

### Review · Reviewer_jsgX · 2025-12-14

**Summary Of Contributions:**

**TL;DR:** The paper shows that fine-tuning a pre-trained lifelong learning model on selectively retrieved samples from a stored dataset can help with performance when adapting to a new task from the same distribution.

The paper studies the problem of test-time adaptation, where an agent is pre-trained sequentially on a set of tasks from expert trajectories (imitation learning), and at test time it is asked to perform a new task coming from the same distribution as the original set of tasks. The agent stores expert trajectories from training tasks on a replay buffer. The paper proposes a method called retrieval-based weighted local adaptation (RWLA) that, at test/deployment time, trains the agent on selectively sampled old experience, weighted by its relevance to the new task. The main contributions of the paper is the distance measure used in this selective retrieval and the mechanism used to weight the contributions of different sections of a trajectory in this local adaptation.

### Strengths:
- The proposed method is quite general and can be used in a plug-and-play manner with a variety of learning methods. For instance, the paper shows this plug-and-play characteristic of RWLA with experience replay (ER) and average gradient episodic memory (AGEM) methods.
- The RWLA mechanism shows substantial improvements in task completion rate over both ER and AGEM methods, across various LIBERO benchmarks.

### Weaknesses:
- The paper heavily uses the phrase “lifelong learning”, however the main experiments are conducted in a single new task setup. That is, after the pre-training step the agent, instead of observing and adapting to a sequence of new tasks, only sees a single new task. After seeing this task, it resets the model, forgetting all new knowledge. As such, the paper does not talk about lifelong learning in the usual sense; rather this is the setting of “test-time adaptation”.
- The paper claims the method is a task-agnostic method (and define the term as meaning non-reliance on both task IDs and task boundaries). However, this method assumes task boundaries both at training and test/deployment time (see Point 1 in requested changes).
- Not enough experimental details are provided, it would be difficult to replicate the results of the paper with the information given.
- The choice of baselines is rather unfair, since the baselines were actually designed for the “full” lifelong learning settings, whereas RWLA is solving a different task (that of test-time adaptation).

**Additional Comments:**

## Questions for the authors
- Is the policy model in Figure 2 the policy model used to also train the baselines?
- What is the difference between “embeddings” and “demo embeddings” in Figure 5?
- Are the baselines also reset, just like how RWLA is reset (as discussed in Sec. A.7.1), in between new tasks during the testing phase?
- In the same vein as “blurry task boundaries” refers to paraphrasing (i.e. it only considers text), is it possible that one could also have vision-based blurry task boundaries (similar scene but very different tasks)? Would that make the task more challenging?

## Aspects which would strengthen the paper (action is appreciated but not required)

- Abstract could more clearly define all the contributions of the method: for instance, it could say that the paper is proposing a plug-and-play method for test-time adaptation, and then quickly describe the three components (data retrieval from the experience replay buffer, selective weighting, and fine-tuning).
- The method is only tested on manipulation tasks, and thus this should be made clear in the title and introduction/motivation (“robot manipulation” instead of “robot learning”).
- page 1, In the third paragraph of introduction, the paper says  “proposed algorithm effectively handles multiple continually encountered tasks and integrates new knowledge without relying on known task boundaries or IDs”. This wording could be changed, since the model doesn’t integrate any knowledge at deployment time, and simply relearns the needed task and then reverts back to the original pre-trained model, effectively discarding the new task’s knowledge.
- page 2: “sequences of open-ended robotics tasks” → what’s “open-ended” about this setting?
- page 3: Just like meta-learning, RWLA also has access to the entire set of tasks (during the pre-training phase), even if it learns them sequentially (this sequentiality is purely artificial, as we are still doing pre-training and not lifelong learning) . So perhaps this could be mentioned.
- page 8: saying that seeds are 1, 21, 42, sounds a bit suspect (sounds like these were cherry picked to have good performance). Perhaps, either a text saying that these were picked arbitrarily could be added. An alternative is to just discard the exact numbers if these are not too important (which it should not be) and relegate this information to the appendix, further mentioning that the specific choice of seeds is arbitrary.
- Page 10, Section 6, there are no real conclusions or discussions here: this section is a copy of the abstract. Perhaps, some nice insights, gained from the experiments, could be added here? Another option is to just change the section heading to "Summary" or "Limitations".
- More emphasis could be placed on the fact that the paper and the algorithm are in the behavior cloning / imitation learning setup (as opposed to control or reinforcement learning like setup).
- Figure placement throughout the text can be improved. For instance, Figures 1 and 2 appear 2-3 pages before they are mentioned in the text. In another instance, Figure 10 is referred before Figure 9. Such a figure (and table) placement can throw the reader off.
- Expand the related works section: some more discussion of selective weighting methods could be added. For instance, one example (which is not really about robotics or behavior cloning) is PER (Schaul et al., 2015).

## Minor comments and typos
- In the abstract: “Besides, task boundaries and IDs are unavailable in scalable, real-world settings, our method enables a lifelong learning robot to perform effectively without relying on such information.” → the grammar could be improved
    - "Besides, task boundaries..." --> "Since task boundaries..."
- In the abstract: ''forgotten'' -> ``forgotten'' (the beginning air quotes change)
- The paper uses too many sub-sub sections --> this might make it difficult for a reader to keep track of things. Perhaps, the paper could go up to only sub-sections (so no sections like 5.1.x)
- The paper uses em-dashes inconsistently. For instance, some sentences have < text > --- < text >, whereas some have < text > --- < text >; and some places use < text > - < text >
- At multiple places in the text, when referring to equations, the paper uses "Eq < number >"; this could be changed to "Eq. < number >"
- page 1: "Noteworthy, task-agnostic does not…" should be changed to "Note that, ..." or "It is noteworthy that ..."
- page 2, Figure 1 caption: perhaps a line about the "Testing" phase could be added
- Typos: “adatped” and “near by” in the “Testing” box inside Figure 1
-  in Figure 1, the text “Subset of Data” has the shadow effect on; not sure if this was intentional
- page 2: “contributions are summarized as” → “contributions are summarized as follows”
- page 3: what is a "task"?
- page 3: define scenario $\mathcal{S}_n^k$; what is $p(\mathcal{T}_k)$
- page 4, Eq. 1: what does $(o_t, a_t) \sim \mathcal{D}_k$ mean? On the earlier page, $\mathcal{D}_k$ was a set, but now it's a distribution? -> Perhaps, this notation could be improved?
- page 4, Eq. 1: the loss function $\mathcal{L}$ is not defined.
- page 5, Algorithm 1, line 1: tasks are given or inputted, not initialized, right?
- Eq. 1 has a sum over $K$s, so line 2a is confusing.
- page 5, section 4.2.1: at what frequency are images sampled from the demonstration?
- page 5: wrong citation for AGEM in section 5.1.2.
- page 7: "task boundaries (See Figure 7)." --> "task boundaries (see Figure 7)."
- page 8: "Noteworthy, this strategy works well early on" --> use something other than noteworthy to start the sentence; check the suggestion given above
- page 9: if space is not too big of an issue, Table 2 could be moved at the top of the page?
- page 9: “Overall ASR” entry in Table 2 could be dropped? (how informative is the average success rate over different lengths of training, anyways?)
- page 9: specify the full form of RULA; is it retrieval-based uniform local adaptation?
- Figure 5a, b are quite interesting; perhaps some quick explanation as to why BERT fails to cluster these tasks could be provided?
- Legend of Figure 5c: if the competing methods could be given more descriptive names, it would be better. For instance, "ASR_er_bert” could become “ER + BERT (ASR)”
- Figure 5c: perhaps this figure could be split into two: one for ASR and one for RA?
- page 11, last line: citation of Ebbinghaus (2013) looks weird: “[image] memory”
- page 13, citation for Phi-3 looks weird. Should it not be something like “Microsoft Phi-3 (2024) …”
- page 16: Appendix A.1 was never referred to in the main text.
- page 16, Table 3: Caption should end in a full stop.
- page 16, Table 3: Using $l_k$ and $l_n$ for two different things is poor notation.
- page 16, Table 3: What are labels in $y_{i, n}$; was this discussed in the main text?
- page 17: font size of tables 4 and 5 should match
- page 17, "Temporal transformer" paragraph: the phrase "and feed the sequence (up to 10 steps)" is unclear; in particular, what does "up to" mean? Could the sequence length be smaller than 10?
- page 17: "demonstration data was initially collected and provided by LIBERO benchmark." --> what does "initially" collected mean? How is it collected now? Also, perhaps write "taken from" instead of "provided by".
- page 17: "demonstrations with the latest version to obtain" --> latest version of what?
- page 17: "It is important to note that occasional rollout failures occurred because different versions of RoboMimic Simulation (Mandlekar et al., 2021) utilize varying versions of the MuJoCo Engine (Todorov et al., 2012)." --> this is highly unclear; perhaps more explanation could be provided?
- page 18, table 6, "Number of Demos per Task" line: perhaps the footnote could be moved in the first column; also, there shouldn't be a space before the "1" of the footnote
- page 18: "Our dataset inherent the dataset" --> is the word inherited?
- page 18: "from LIBERO Liu et al. (2024)" --> use \citep here
- page 18: "Additionally, we add demo" --> use “added”
- page 18: "Phi-3-mini-4k-instruct model (mini-4k instruct, 2024)" --> perhaps the citation should have "Microsoft" instead of "mini-4k instruct"?
- page 18: "both BERT and Sentence Similarity Model struggle to distinguish" --> didn't Figure 5b show that the Sentence Similarity model could distinguish the tasks very well?
- page 19: "A row-wise minimum over the EDM" --> is it a row-wise or column-wise minimum?
- page 20: “used in Eq. equation 3.” → “used in Eq. 3.”

**Audience:**

Yes

**Audience Explanation:**

Yes, test-time adaptation is likely to be of interest to some members of TMLR's audience. With the advent of general purpose models that have been pre-trained on a large number of tasks (such as GATO, VLAs, and chatbots like ChatGPT), test-time local adaptation methods will become increasingly important. For instance, practitioners could pick one of these pre-trained models and then selectively adapt these models on the tasks that they care about. This paper’s proposed algorithm has the potential to speed up this adaptation process, while improving performance, when compared to existing local adaptation approaches (such as naive experience replay methods). Furthermore, because of the plug-and-play characteristics of this method, it could potentially be employed in a variety of related areas, if its performance is upheld. Finally, the weighting mechanism proposed in the paper could be of independent interest to ML practitioners.

**Broader Impact Concerns:**

The paper doesn’t provide a broader impact statement present.  Some potential concerns are:
- The current method has possible interpretability and safety issues, especially since the RWLA algorithm adapts robotic systems during test-time.
- Replay based methods have inherent data privacy concerns.

However, I acknowledge that issues like these are present in many ML algorithms and perhaps don’t necessarily warrant a special mention in this paper.

**Claims And Evidence:**

No

**Claims Explanation:**

### **Claim 1: RWLA enables a lifelong learning robot to perform effectively without relying on task boundaries and task IDs.**

**Lifelong learning claim:** The setting considered in the paper is that of “test-time local adaptation”, not of lifelong learning. The paper does use a lifelong learning setting in the pre-training phase (see Alg. 1), however, the pre-training part is not the novel contribution. The novelty of the paper lies in the “Reviewing phase” (see Alg. 1). But in this phase, the paper effectively only considers a single new task. I say this because, as outlined in Section A.7.1, after each new task, the model is reset to the preserved stable model from the pre-training phase. Consequently, the agent effectively only sees a single new task. Ultimately, the paper simply presents a method to “relearn at deployment time what was forgotten during training” for a specific task.

**''Not relying on task boundaries'' claim:** An argument can be made against this claim in three ways:
- During pre-training, the agent is trained on a sequence of tasks with clear boundaries.
- Since at deployment time the agent effectively only sees a single new task, the very notion of task boundaries is irrelevant.
- Even if we argue that the agent does see a sequence of new tasks at deployment time, its model is always reset to the original pre-trained model after each task. And in order to do so, the agent must have access to some way of knowing when one task ends and the next one begins, i.e. it relies on task boundaries.

### **Claim 2: Experimental results across diverse manipulation tasks demonstrate that our framework provides a plug-and-play paradigm for lifelong learning, enhancing robot performance in open-ended, task-agnostic scenarios.**

I am not fully convinced by the experimental setup and analysis of this paper (see Point 4 in requested changes). Firstly, the baselines chosen for the paper should be stronger. For instance, RLWA is not compared to methods designed for test-time adaptation (such as L2M, TAIL), but is instead only compared to lifelong learning methods. This can be unfair since lifelong learning methods need to balance between learning new skills while not forgetting old ones. In contrast, test-time methods need only care about learning new skills. As a result, lifelong learning methods are solving a different (and harder) problem. Secondly, the comparison with Packnet is also flawed.

Finally, we point out some empirical analysis issues: all the experiments used 3 seeds, and thus the results might not be statistically significant. In addition to this, there was no discussion of how fair the hyperparameter selection process was for all the methods. (We would like to point the authors to the papers by Patterson et al. 2024 and Henderson et al. 2018 for more discussion on these points.)

L2M: Schmied et al. (2023). Learning to Modulate pre-trained Models in RL. NeurIPS 2023

TAIL: Liu et al. (2024). Task-specific Adapters for Imitation Learning with Large Pretrained Models. ICLR 2024.

Patterson et al. (2024). Empirical design in reinforcement learning. JMLR.

Henderson et al. (2018). Deep reinforcement learning that matters. AAAI.

**Requested Changes:**

I write the major issues here. Questions for the authors are given in ''Additional comments'' (answer is expected). Furthermore, changes which would strengthen the work but for which action is not required/expected, as well as minor issues, are also given in the ''Additional comments'' section.

## 1. Improving the paper’s focus and scope:

1. Remove the whole idea of “lifelong learning”, and phrase this as a novel method for test-time adaptation.
    - For instance, STRAP (Memmel et al. 2024) operates in a very similar problem setting as this paper, while positioning itself as an approach for few-shot robotic manipulation.
    - In a similar vein, perhaps discussions on “stability and plasticity” ought to be removed. For instance, the model remains plastic at deployment time because we reset it after each task; but since we reset it, we forget all the knowledge learnt from the deployment task.
    - The mentioning of “blurry task boundaries” could also be modified, since “blurry” just refers to paraphrasing of the task description by an LLM, and as previously discussed the method still relies on clear boundaries.
2. Reconsider claiming that this method does not rely on task-boundaries. It is okay to say that this method is task-agnostic (meaning it does not rely on task IDs), but the paper defines task-agnostic to mean it also does not rely on task-boundaries (last paragraph page 1), which is not the case for this method.

3. Expand the limitations section:
    - There is no mention of the computational requirements of such a method. In particular, there is no mention of replay buffer sizes. As the method is obviously heavily dependent on the buffer’s data diversity, such numbers could be reported. It could also be important to study how replay buffer size per task impacts RLWA’s performance.
    - This set-up has one major limitation: we require the robot to interact with its deployment environment so we can collect ``failed rollouts’’ and only then correct them. This can be dangerous to do in real environments (deploying a model which has forgotten knowledge might just result in costly failures). This limitation could be acknowledged in the text.
    - Table 2 shows that extra fine-tuning epochs do not correlate with better performance. Thus, it is unclear for how long to fine-tune RWLA. This limitation could be discussed in the paper.

Memmel et al. (2024). Strap: Robot sub-trajectory retrieval for augmented policy learning. arXiv preprint arXiv:2412.15182.

Schaul, T., Quan, J., Antonoglou, I., & Silver, D. (2015). Prioritized experience replay. arXiv preprint arXiv:1511.05952.

## 2. Giving sufficient details about the problem setting

The preliminaries section could be expanded to properly define the problem setting.
- For example, what is a task, what does it mean to succeed or fail in a task, what are environmental settings and language descriptions, etc.
- This could also be expanded to discuss the LIBERO benchmark

## 3. Giving sufficient details about the algorithm and associated methodology:

- page 21: A.7.1 "Crucially, we always return to the preserved final model for subsequent scenarios" should be mentioned in the main paper. This is not quite lifelong learning and this point must be moved to the main text.
- Section A.4 (selective weighting) should be moved to the main paper, as it is a main contribution of the paper. In particular, if a clear pseudocode can be given for it, that would be great. In its current form, it is difficult to figure out exactly how this weighting is obtained.
- Algorithm 1 needs to show that there is a “failing test” based on the 10 rollouts in the deployment task, and that it is based on that test that the algorithm decides whether to perform the reviewing step.  There is also currently no definition as to what constitutes “failure” in this case.


## 4. Experimental details and resulting claims:
- Add baselines that do test-time adaptation. (such as L2M, Schmied et al., 2023; or TAIL, Liu et al., 2023)
- The comparison with PackNet in ''libero_different_scenes'' (Table 1) is possibly flawed. Packnet’s network size is chosen before training to accommodate for the number of tasks it needs to learn. ''Libero_different_scenes'' has 90 tasks. Instead of properly adapting network size for such a setting, the paper keeps the network size to accommodate for 8 tasks only. Thus, while Packnet performs well in initial tasks, it quickly runs out of capacity and simply gets ''0’’ in all following tasks. This can be seen in Table 7 (appendix). Instead, Packnet’s size could be adapted so its performance can be fairly compared.
- The paper states that claim 3 (“selective weighting based on rollout failures is effective”)  is supported. However, Table 2 shows that improvements from using selective weighting are not necessarily significant, and probably are not enough to support claim 3. The improvement is incredibly small at times, and further, averaging over 3 seeds and testing on 20 episodes, doesn’t provide much statistical certainty.
- Please do not compute and report standard deviation from just 3 seeds (e.g.: don't give error bars in Figures 5c and 10). In many ML tasks, 3 seeds are just not enough to estimate standard deviation. (For more discussion on these, please see the papers by Patterson et al. 2024 and Henderson et al. 2018, I mentioned before.)

Schmied et al. (2023). Learning to modulate pre-trained models in RL. NeurIPS.
Liu et al. (2023). Tail: Task-specific adapters for imitation learning with large pretrained models. arXiv preprint arXiv:2310.05905.

---

### Note · Authors · 2026-01-10

I have read and agree with the venue's withdrawal policy on behalf of myself and my co-authors.